# Blockade of the M1 muscarinic acetylcholine receptors impairs eyeblink serial feature-positive discrimination learning in mice

**Md Ashrafur Rahman**[1,2]*, **Norifumi Tanaka**[1], **Md. Nuruzzaman**[2], **Shandhya DebNath**[2], **Shigenori Kawahara**[1,3]

**1** Graduate School of Innovative Life Science, University of Toyama, Gofuku, Toyama, Japan, **2** Department of Pharmaceutical Sciences, School of Health & Life Sciences, North South University, Dhaka, Bangladesh, **3** Graduate School of Science and Engineering, University of Toyama, Gofuku, Toyama, Japan

* rahman.ashrafur@northsouth.edu

**Data Availability Statement:** All relevant data are within the manuscript.

**Funding:** The author(s) received no specific funding for this work.

## Abstract

The serial feature-positive discrimination task requires the subjects to respond differentially to the identical stimulus depending on the temporal context given by a preceding cue stimulus. In the present study, we examined the involvement of the M1 muscarinic acetylcholine receptors using a selective M1 antagonist VU0255035 in the serial feature-positive discrimination task of eyeblink conditioning in mice. In this task, mice received a 2-s light stimulus as the conditional cue 5 or 6 s before the presentation of a 350-ms tone conditioned stimulus (CS) paired with a 100-ms peri-orbital electrical shock (cued trials), while they did not receive the cue before the presentation of the CS alone (non-cued trials). Each day mice randomly received 30 cued and 30 non-cued trials. We found that VU0255035 impaired acquisition of the conditional discrimination as well as the overall acquisition of the conditioned response (CR) and diminished the difference in onset latency of the CR between the cued and non-cued trials. VU0255035 administration to the control mice after sufficient learning did not impair the pre-acquired conditional discrimination or the CR expression itself. These effects of VU0255035 were almost similar to those with the scopolamine in our previous study, suggesting that among the several types of muscarinic acetylcholine receptors, the M1 receptors may play an important role in the acquisition of the conditional discrimination memory but not in mediating the discrimination itself after the memory had formed in the eyeblink serial feature-positive discrimination learning.

## 1. Introduction

The muscarinic acetylcholine receptors (mAChRs) are involved in a broad range of brain functions, such as mnemonic, attentional, and cognitive processes [1]. Among a variety of learning paradigms, the hippocampus-dependent tasks such as spatial learning [2], contextual fear conditioning [3], and trace eyeblink conditioning [4] are especially susceptible to pharmacological manipulation of the mAChRs. This is consistent with the fact that the hippocampus receives

**Competing interests:** have no competing interests exist.

cholinergic inputs originating from the medial septum [5] and the mAChRs are abundant in the hippocampus [6]. Because the classical eyeblink conditioning has several similar paradigms that differ in the degree of hippocampus-dependency [7], it will provide a good model to investigate the roles of the mAChRs in various processes performed in the brain during learning as well as expression of the acquired memory.

In eyeblink conditioning, the pharmacological blockade of the mAChRs severely impaired acquisition of the hippocampus-dependent trace paradigm in rabbits [4] and mice [8], while in the delay paradigm that does not require the intact hippocampus [9–13] it only slowed the learning rate in the delay paradigm in rabbits [4, 14] or did not impair the delay paradigm in mice [15]. Interestingly, the cerebellum-deficient mutant mice that require an intact hippocampus to learn a delay paradigm [16] showed a severe impairment with systemic administration of scopolamine [17].

We recently found that the serial feature-positive discrimination task in eyeblink conditioning, in which the hippocampal theta oscillation might play an important role [18], also depends largely on the mAChRs in mice [19]. The systemic administration of mAChR antagonist, scopolamine, selectively impaired acquisition of the memory for discrimination, but not the expression of the pre-acquired discriminative memory [19]. Although similar kinds of conditional discriminations tasks have been studied in rabbits [20–22] and rats [18], the detail of the involvement of the mAChRs in mice has not been investigated so far. Because amnesic patients also showed severe impairment in the serial conditional discrimination paradigm of eyeblink conditioning [23, 24], it is worthwhile to reveal the detail.

It is well known that the mAChRs are classified into five subtypes, termed as M1–M5 [25]. Among them, The M1 receptors are most abundantly expressed in the hippocampus, constituting 40–50% of the total mAChRs [26–29]. Studies showed that the M1 receptors are more related to memory functions compared to other mAChRs receptors [30, 31]. Besides, the M1 receptors are important for attention and memory in several learning tasks, such as passive avoidance [32, 33], contextual fear conditioning [34], radial arm maze [35], T-maze [36] and considered as a potential target for memory functions [37]. In the hippocampus-dependent trace eyeblink conditioning, it was shown that selective activation of the M1 receptors improved the memory in aged rabbits by enhancing the excitability of hippocampal pyramidal neurons [38]. Consistent with this, an electrophysiological study using mAChRs knock-out mice revealed that the M1 but not the M3 receptors are involved in the cholinergic enhancement of hippocampal long-term potentiation [39]. Immunohistochemical study also showed a selective increase in immunoreactivity of PKCγ isoform after trace eyeblink conditioning, suggesting the involvement of signaling pathways of M1 receptors [40]. In contrast to the abundance of M1 receptors in the pyramidal neurons [41] which are highly engaged in learning the hippocampus-dependent eyeblink [38], the M2 receptors are present only in the nonpyramidal neuron in the cortex and hippocampus [27]. In addition, they are densely expressed on GABAergic interneurons [42]. They play a vital role in the inhibitory modulation at dopaminergic terminals [41, 43, 44] as well as in a general anti-nociception at the spinal cord [45] but have a minor role in learning and memory compared to the M1 receptors [46]. Consistent with this, a significant correlation was found between the performance in spatial learning and the M1 receptor binding, but not the M2, in the hippocampus of aged monkeys [47]. The M3 receptors are found in the brain but at a lower level than other subtypes. They are mostly involved in regulating food intake [48] and body growth [49]. The M4 receptors are largely expressed in the corpus striatum and considered as a promising target for treating schizophrenia [50] and locomotor dysfunction such as Parkinson's disease [51, 52]. The M5 receptors are predominantly expressed in the pars compacta of substantia nigra and are the potential target for the treatment of drug addiction [53]. Therefore, we focused on the M1 receptors as the first

candidate that plays an important role in the serial feature-positive discrimination task of eyeblink conditioning, although the other subtypes of mAChRs that constitute 50–60% of the hippocampal mAChRs might serve some roles in this learning as well.

In the previous study, we examined the roles of (mAChRs) in the serial feature-positive discrimination task of eyeblink conditioning in mice, using a non-selective mAChR antagonist scopolamine. Some studies showed that a selective M1 antagonist, VU0255035, produced a similar antidepressant response to the action of scopolamine through the medial prefrontal cortex [54, 55]. Therefore, it is likely that the M1 selective antagonist might impair the formation of the memory for serial feature-positive discrimination in eyeblink conditioning as well as scopolamine, although the established discriminative performance was not impaired by scopolamine [19].

In the present study, we investigated the roles of the M1 antagonist, in the serial feature-positive discrimination task in mouse eyeblink conditioning by using VU0255035, which has excellent brain penetration and a greater than 75-fold selectivity for the M1 receptors over other subtypes of mAChRs [56]. We found that systemic administration of VU0255035 impaired the acquisition of conditional discrimination, but did not disturb the performance of pre-acquired conditional discrimination, similarly to the effects of non-selective antagonist scopolamine in our previous work. These results suggested that the M1 receptors play an important role that could not be fully compensated by other subtypes during the formation of memory for conditional discrimination.

## 2. Materials and methods

### 2.1. Animals and ethics statement

Sixteen male 8-week-old C57Bl/6 mice were purchased from Japan SLC, Inc. (Hamamatsu, Shizuoka, Japan) and individually housed in standard plastic cages under a 12/12-h light/dark cycle at $24 \pm 2°C$ with ad libitum access to food and water. The experiments were performed during the light period. All the experimental procedures were conducted following the NIH Guide for the Care and Use of Laboratory Animals and approved by the Experimental Animal Committee of the University of Toyama. Throughout the experiments, all efforts were made to minimize the number of animals used and to optimize their comfort.

### 2.2. Surgical procedures

The surgical procedures were the same as in our previous study [19]. In brief, mice were anesthetized with ketamine (80 mg/kg, i.p.; Sankyo, Tokyo, Japan) and xylazine (20 mg/kg, i.p.; Bayer, Tokyo, Japan). During surgery, isoflurane (1–2%, Abbot Japan, Osaka, Japan) was used when necessary. Four Teflon-coated stainless-steel wires (140 μm in diameter, A-M Systems, Sequim, WA, USA) were subcutaneously implanted in the left upper eyelid. Two of them were used to record EMGs for the CR detection and the remaining two to deliver electrical shocks as the US. The connector pins soldered to the wires were fixed to the skull with the help of dental acrylic resin and stainless steel screws. Subsequently, mice were injected with ampicillin (100 mg/kg, i.p.; Meiji Seika, Tokyo, Japan), placed in a warm cage until they moved voluntarily, and then returned to their home cage.

### 2.3. Drugs

The mice were blindly divided into the control and VU0255035 groups (n = 8 each). VU0255035 (Tocris Bioscience, Ellisville, MO, USA) was dissolved in the saline solution containing 5% DMSO (Wako, Osaka, Japan) and administered intraperitoneally at a final dose of

10 mg/kg [56] in a volume of 5 mL/kg. The control group received an equivalent volume of the DMSO-containing saline solution.

## 2.4. Conditioning procedure

The conditioning apparatus and procedures were the same as in our previous study [19]. After at least three days of recovery from the surgery, each mouse was placed into an acrylic cylinder, connected to a lightweight cable, and allowed to acclimate to the experimental apparatus (spontaneous session 1). The next day, mice received an intraperitoneal injection of the vehicle (5% DMSO) or VU0255035 15 min before the start of the session (spontaneous session 2). Mice were then conditioned in the serial feature-positive discrimination task with the daily administration of the vehicle or VU0255035 for 7 days (acquisition session 1–7). The conditioning was continued for additional three consecutive days (expression sessions 1–3) by switching the injected solution from the vehicle to VU0255035 in the control group and from VU0255035 to the vehicle in the VU0255035 group.

Each daily conditioning session consisted of a random sequence of 30 cued and 30 noncued trials separated by an intertrial interval randomized between 60 and 70 s. The same type of trials did not repeat consecutively more than twice. In the cued trial, a sequence of the 2-s light cue, 3- or 4-s random delay period, 350-ms tone CS (1 kHz, 85–87 dB, rise and fall times of 5 ms), and 100-ms periorbital electrical shock US (100 Hz) co-terminated with the CS was presented (Fig 1A). The light cue was delivered by five green light-emitting diodes (3 cd each) on the sidewall of the chamber and the tone CS by a speaker over the cylinder. The US intensity started from 0.3 mA and was adjusted individually and daily for each mouse to elicit an eyeblink/head-turn response, which was monitored with an infrared camera. In the non-cued trial, the CS was presented alone without the US or the preceding light cue.

## 2.5. EMG analysis

The eyelid EMG was band-pass filtered between 0.15 and 1.0 kHz, sampled at 10 kHz, and analyzed using a custom-made program written in MATLAB (Mathworks, Natick, MA, USA) in the same way as in our previous study [19]. In brief, EMG signals were extracted by setting a cutoff threshold at the level of mean ± standard deviation over the data of 300-ms period before the CSs in a session and integrated with a 20-ms decay time constant. The integrated EMG signal was considered significant if it exceeded 30% of the threshold. Each trial was

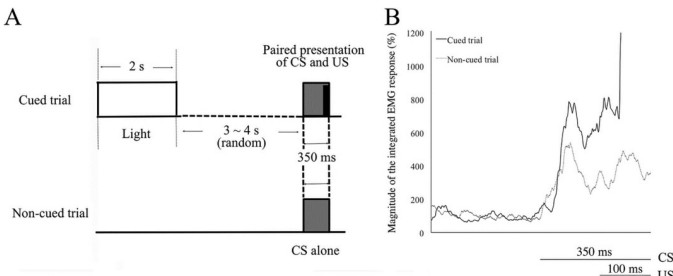

**Fig 1. Eyeblink serial feature-positive discrimination task in mice.** (A) The schematic time sequence of the cue, the conditioned stimulus (CS), and the unconditioned stimulus (US). Thirty cued and non-cued trials were randomly performed in a daily session with intertrial intervals between 60 and 70 s. In a cued trial, a 2-s light cue was delivered randomly 5 or 6 s before a 350-ms tone CS, which co-terminated with a 100-ms periorbital electrical shock US. In a non-cued trial, the CS was presented alone without the preceding cue or the US. (B) Examples of integrated EMG of a mouse averaged over the cued (solid line) and non-cued trials (dotted line) in a session. The horizontal axis represents the time course (ms) and the vertical axis illustrates the magnitude of the EMG response (%).

judged as valid if it did not show significant EMG signals in more than 30% of the 200-ms pre-CS period. Among the valid trials, a trial was considered to contain the CR if it showed significant EMG signals in more than 30% of the 200-ms pre-US period. We calculated the CR% in each trial type by dividing the number of CR-containing trials by that of valid trials in each trial type of the daily session, the differential CR% by subtracting the CR% in non-cued trials from that in cued trials, and the CR% for overall acquisition by dividing the sum of the number of all CR-containing trials by that of all valid trials in the daily session. All of them were expressed as a percentage. The averaged EMG amplitude was calculated by averaging the integrated EMG amplitude over the first 250 ms of the CS and expressed as a percentage of that averaged over 300 ms before the CS onset. Therefore, the averaged EMG amplitude during the CS does not depend on the CR judgment. Also, the latencies from the CS onset to the time when the integrated EMG signal exceeded 30% of the threshold for the first time (onset latency) and to the peak of integrated EMG signal (peak latency) were calculated as well as the peak amplitude in each CR-containing trial.

## 2.6. Statistical analyses

Statistical analyses were carried out using the SPSS statistical software (SPSS, Chicago, IL, USA). Data were expressed as the mean ± standard error of the mean (SEM). Using the two-way repeated-measures ANOVA or the paired *t*-test, differences in measured values were analyzed. Differences were considered statistically significant when the p-value was less than 0.05.

# 3. Results

## 3.1. Effects on the acquisition of the discriminative conditioned response

In the serial feature-positive discrimination, animals received two kinds of randomly alternating trials, cued reinforced trials and non-cued non-reinforced trials (Fig 1A). In the cued trials mice received a serial presentation of a feature conditional cue of green light and the conditioned stimulus (CS) of 1-kHz tone with a random temporal gap between them, followed by the unconditioned stimulus (US) of periorbital shock, while in the non-cued trials mice received only the CS without the preceding conditional cue. The conditioned responses (CRs) to the CS was monitored by recording the eyelid electromyograms (EMGs) (Fig 1B). Mice received an intraperitoneal injection of the vehicle (5 ml/kg of 5% dimethyl sulfoxide; DMSO) or VU0255035 (10 mg/kg) before the daily conditioning session.

Consistent with our previous results [19] the vehicle-injected control mice successfully learned to display a much higher number of CRs in cued trials than in non-cued trials (Fig 2Aa), despite the identical CS in both types of trials. The frequency of CR occurrence (CR%) reached around 50% in cued trials ($50.3 \pm 5.3\%$ on the last day of acquisition), whereas it remained low around 20% in non-cued trials ($18.5 \pm 3.5\%$ on the last day of acquisition). Two-way repeated-measures analysis of variance (ANOVA) demonstrated that mice acquired the discrimination between cued and non-cued trials (session $F_{(8, 56)} = 7.99$, $p < 0.001$, trial-type $F_{(1, 7)} = 91.2$, $p < 0.001$, session × trial-type interaction $F_{(8, 56)} = 8.6$, $p < 0.001$). We also confirmed the successful discrimination between the cued and non-cued context by examining the EMG amplitude averaged over the first 250 ms of CS in all valid trials of each type, irrespective of the CR occurrence. The EMG amplitude was expressed as a percentage of that averaged over 300 ms before the CS (Fig 3A). The average EMG amplitude increased to around 700% in cued trials during 7 days of conditioning (Fig 3Aa; $717.2 \pm 154.7\%$ in the last acquisition day), whereas it remained low around 250% in non-cued trials (Fig 3Aa; $277.7 \pm 87.4\%$ in the last acquisition day). Two-way repeated-measures ANOVA of the average EMG amplitude revealed significant effects of session ($F_{(8, 56)} = 4.96$, $p < 0.05$) and trial type ($F_{(1, 7)} = 78.91$,

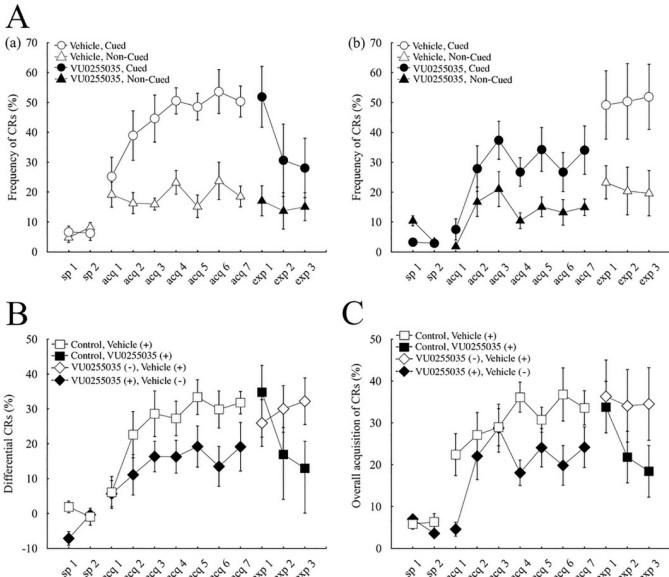

**Fig 2. Effects on the occurrence of the conditioned response (CR).** (A) The frequency of the CR occurrence (CR%) in cued (circles) and non-cued (triangles) trials in vehicle-injected control (a) and VU0255035-treated (b) groups. After two days of adaptation sessions (sp1–2), 8 mice of each group underwent 7 days of conditioning for the acquisition of the conditional CRs (acq1–7), followed by three days of conditioning with an injection of an exchanged solution to test the effects on expression of the acquired CR (exp1–3). Empty and closed symbols denote the administration of the vehicle and VU0255035, respectively. (B, C) The differential CR% calculated by subtracting the CR% in non-cued trials from that in cued trials (B) and the overall CR% calculated from the total number of CRs during the session irrespective of the trial types (C) in vehicle-injected control (squares) and VU0255035-treated (diamonds) groups. Empty and closed symbols indicate the administration of the vehicle (marked as vehicle (+)) and VU0255035 (denoted as VU0255035 (+)), respectively. The vertical bars indicate the standard error of the mean.

$p < 0.001$), and a significant interaction between session and trial type (F(8, 56) = 10.11, $p < 0.001$).

Compared to the control mice, VU0255035-injected mice showed moderate learning: the CR% reached the plateau at around 30% in cued trials (34.0 ± 8.2% in the last day of acquisition), whereas it remained around 15% in non-cued trials (14.9 ± 2.8% in the last day of acquisition) (Fig 2Ab). A statistical comparison using two-way repeated-measures ANOVA revealed significant effects of session (F(8, 56) = 5.28, $p < 0.001$) and trial type (F(1,7) = 33.67, $p < 0.01$), as well as a significant interaction between session and trial type (F(8,56) = 3.59, $p < 0.05$). Analysis of the EMG amplitude averaged over the first 250 ms of CS also confirmed

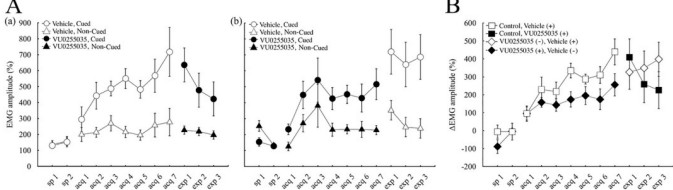

**Fig 3. Effects on the EMG amplitudes during the CS.** (A) The averaged EMG amplitude over the first 250 ms of CS, expressed as a percentage of that averaged over 300 ms before the CS onset, in cued (circles) and non-cued (triangles) trials in (a) vehicle-injected control (n = 8) and (b) VU0255035-treated (n = 8) groups. (B) The differences in the averaged EMG amplitudes between cued and non-cued trials in control (squares) and VU0255035-treated (diamonds) groups. Empty and closed symbols represent the administration of the vehicle (denoted as vehicle (+)) and VU0255035 (marked as VU0255035 (+)), respectively. The vertical bars indicate the standard error of the mean.

the moderate learning in VU0255035-injected mice (Fig 3Ab). The average EMG amplitude increased to around 450% in cued trials (458.0 ± 87.0% in the last acquisition day), whereas it remained low around 200% in non-cued trials (202.1 ± 26.6% in the last acquisition day). Two-way repeated-measures ANOVA of the average EMG amplitude revealed a significant effect of trial type ($F_{(1,7)}$ = 36.75, $p < 0.05$) and a significant interaction between session and trial type ($F_{(8,56)}$ = 6.73, $p < 0.001$), but no significant effect of session ($F_{(8,56)}$ = 2.94, $p > 0.05$).

To compare the acquisition of discrimination in the drug-treated mice with that in control mice, we subtracted the CR% value in non-cued trials from that in cued trials in each mouse (Fig 2B). Both groups of mice developed differential CRs, but the degree of discrimination was much lower in VU0255035-injected mice than in control mice. The differential CR% reached around 30% in control mice, whereas it remained low around 15% in VU0255035-treated mice. Two-way repeated-measures ANOVA revealed significant effects of session ($F_{(8,56)}$ = 10.19, $p < 0.001$) and group ($F_{(1,7)}$ = 18.58, $p < 0.05$), but no interaction between session and group ($F_{(8,56)}$ = 0.85, $p > 0.05$), indicating that VU0255035 impaired acquisition of the discrimination between the cued and non-cued contexts. Analysis of the difference in the EMG amplitude averaged over the first 250 ms of CS also confirmed the inhibitory effect of VU0255035 on the acquisition of discrimination (Fig 3B). The differential EMG amplitude increased to around 450% in control mice during 7 days of conditioning, whereas it remained low around 250% in VU0255035-treated mice. Two-way repeated-measures ANOVA revealed significant effects of session ($F_{(8,56)}$ = 15.81, $p < 0.001$) and group ($F_{(1,7)}$ = 6.49, $p < 0.05$), but no significant interaction between session and trial type ($F_{(8,56)}$ = 1.05, $p > 0.05$).

To further examine the effects of VU0255035, the overall frequency of CR occurrence was calculated by combining the number of CR trials in the cued and non-cued trials (Fig 2C). We found that VU0255035-injected mice developed a fewer number of overall CRs than control mice. The CR% for the overall acquisition was around 35% in control mice, whereas it remained low around 20% in VU0255035-treated mice. Two-way repeated-measures ANOVA revealed significant effects of session ($F_{(8,56)}$ = 10.74, $p < 0.001$) and group ($F_{(1,7)}$ = 10.32, $p < 0.05$), but no interaction between session and group ($F_{(8,56)}$ = 1.75, $p > 0.05$).

### 3.2. Effects on expression of the pre-acquired conditioned response

After 7 days of conditioning, the injected solution was changed from the vehicle to VU0255035 in control mice to investigate possible effects of VU0255035 on the expression of the pre-acquired CR. As shown in Fig 2Aa, VU0255035 did not impair the expression of the pre-acquired CR in the cued trials as well as in the non-cued trials. The CR% in the last acquisition session and the first expression session was 50.3 ± 5.3 and 51.8 ± 10.2%, respectively, in cued trials ($p > 0.05$, paired $t$-test) and 18.5 ± 3.5 and 17.0 ± 5.1%, respectively, in non-cued trials ($p > 0.05$, paired $t$-test). Consistent with the effect of VU0255035 during acquisition sessions in the drug-treated mice, the CR% in the cued-trials decreased during the three consecutive expression sessions in control mice (F = 8.68, $p < 0.05$, one-way ANOVA and post hoc Tukey's test, $p < 0.05$) and became comparable to those in the late acquisition sessions in the VU0255035-treated mice. These effects of VU0255035 were also confirmed by the analysis of differential CR% (Fig 2B), overall CR% (Fig 2C), average EMG amplitude (Fig 3Aa), and differential EMG amplitude (Fig 3B). All of these parameters were not significantly changed by the administration of VU0255035 in the first expression session ($p > 0.05$ each, paired $t$-test) but showed a tendency to decrease during the three consecutive expression sessions in control mice.

In contrast, when the injected solution was changed from VU0255035 to the vehicle in VU0255035-treated mice, both the CR% in cued and non-cued trials showed a tendency to increase, although there was no significant difference between the last acquisition session and the first expression session (p > 0.05, paired *t*-test). Besides, CR% in cued-trials remained high during the three expression sessions with vehicle injection (Fig 2Ab), which was comparable to those in the late acquisition sessions in control mice. The average CR% in cued trials during the last three acquisition sessions and the three expression sessions was 31.6 ± 5.2 and 50.4 ± 10.3%, respectively (p < 0.05, paired *t*-test), whereas that in non-cued trials, was 14.4 ± 2.1 and 21.1 ± 5.2%, respectively, and showed no significant difference (p > 0.05, paired *t*-test). These findings were also confirmed by the analysis of differential CR% (Fig 2B), overall CR% (Fig 2C), average EMG amplitude (Fig 3Ab), and differential EMG amplitude (Fig 3B). All of these parameters showed a tendency to increase in the first expression session compared to the last acquisition session (p > 0.05 each, paired *t*-test) and kept a high level during the subsequent sessions with vehicle injection in VU0255035-treated mice.

### 3.3. Effects on the temporal pattern of conditioned response

To investigate the effects of VU0255035 on the temporal pattern of CR, the latency from the CS onset to the CR onset (onset latency) and that to the CR peak (peak latency), as well as the amplitude of the CR peak (peak amplitude), were examined (see details in materials and methods).

Consistent with our previous results [19], the onset latency of CR in control mice was shorter in cued trials than in non-cued trials during the latter half of the acquisition sessions (Fig 4A). The average onset latency over the last 4 days of acquisition sessions in cued and non-cued trials was 91.5 ± 3.8 and 112.7 ± 5.5 ms, respectively, and showed a significant difference between the trial types (p < 0.01, paired *t*-test). In contrast, the difference in the onset latency tended to be small in VU0255035-injected mice (Fig 5A). The average onset latency over the last 4 days of acquisition sessions in cued and non-cued trials was 96.2 ± 4.8 and 104.6 ± 6.3 ms, respectively, and showed no significant difference between the trial types (p > 0.1, paired *t*-test).

In contrast to the onset latency of CR, the peak latency (Fig 4B) and the peak amplitude (Fig 4C) were similar between the cued and non-cued trials in control mice, as reported in our previous results [19]. The average peak latency over the last 4 days of acquisition sessions in cued and non-cued trials was 147.3 ± 5.2 and 154.0 ± 7.3 ms, respectively, and showed no significant difference (p > 0.05, paired *t*-test). The average peak amplitude over the last 4 days of acquisition sessions in cued and non-cued trials was 76.5 ± 2.8 and 70.1 ± 3.9%, respectively, and showed no significant difference (p > 0.05, paired *t*-test). The VU0255035-

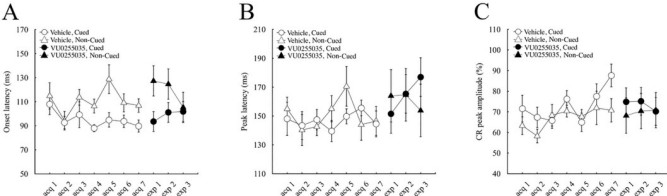

**Fig 4. Change of the temporal parameters of CR in the control group.** Onset latency (A), peak latency (B), and peak amplitude (C) of CR in cued (circles) and non-cued (triangles) trials of vehicle-injected control group mice (n = 8). Empty and closed symbols illustrate the administration of the vehicle and VU0255035, respectively. The vertical bars indicate the standard error of the mean.

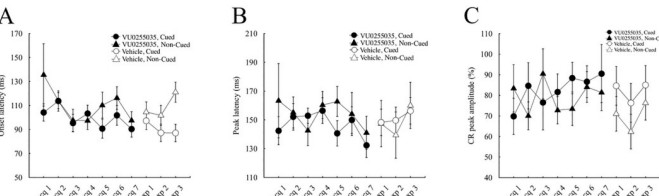

**Fig 5. Change of the temporal parameters of CR in the VU0255035-administered group.** Onset latency (A), peak latency (B), and peak amplitude (C) of CR in cued (circles) and non-cued (triangles) trials of VU0255035-treated mice group (n = 8). Empty and closed symbols represent the administration of the vehicle and VU0255035, respectively. The vertical bars indicate the standard error of the mean.

injected mice also showed no difference in the peak latency and peak amplitude between cued and non-cued trials (Fig 5B and 5C). The average peak latency over the last 4 days of acquisition sessions in cued and non-cued trials was 144.8 ± 5.9 ms and 155.7 ± 7.7 ms, respectively, and showed no significant difference (p > 0.05, paired *t*-test). The average peak amplitude over the last 4 days of acquisition sessions in cued and non-cued trials was 85.9 ± 5.4 and 76.7 ± 4.3%, respectively, and showed no significant difference (p > 0.05, paired *t*-test). Note that there were no differences in the peak amplitude between the control and the VU0255035-injected mice (Figs 4C and 5C), indicating no direct effect of VU0255035 on the expressed CRs.

### 3.4. Effects on the temporal pattern of the pre-acquired conditioned response

Further analysis of the temporal pattern of CR (Fig 4A) revealed that VU0255035 administration after sufficient learning in control mice did not significantly change the onset latency in cued trials: 89.6 ± 5.2 ms and 93.5 ± 7.2 ms in the last acquisition session and the first expression session, respectively (p > 0.05, paired *t*-test), while it increased the onset latency in non-cued trials from 106.7 ± 5.7 ms to 127.2 ± 8.9 ms (p < 0.05, paired *t*-test). Consistent with the effect of VU0255035 during acquisition sessions in the drug-treated mice (Fig 5A), the significant difference in the onset latency of CR between cued and non-cued trials in the first expression session (p < 0.05, paired *t*-test) was diminished by conditioning with VU0255035 administration during the expression sessions in control mice.

In VU0255035-treated mice, the change of the injected solution from VU0255035 to the vehicle did not significantly alter the onset latency of CR in cued and non-cued trials (Fig 5A; p > 0.05 each, paired *t*-test): 90.3 ± 6.8 ms and 97.2 ± 4.6 ms in the cued trial of the last acquisition and the first expression sessions, respectively, and 97.5 ± 7.5 ms and 104.5 ± 6.4 ms in the non-cued trial. In quite a contrast to the onset latency of CR during the expression sessions in control mice (Fig 4A), a difference in the onset latency between the trial types developed during the three consecutive expression sessions (p < 0.05, paired *t*-test in the third expression session).

In both treatment groups, there were no significant differences in the peak latency and the peak amplitude between the last acquisition session and the first expression session in both cued and non-cued trials (Figs 4B, 4C, 5B and 5C; paired *t*-test, p > 0.05).

## 4. Discussion

In the present study, we found that the M1 mAChR antagonist VU0255035 impaired acquisition of the differential CRs between the cued and non-cued trials as well as the overall CRs during the serial feature-positive discrimination task of eyeblink conditioning. However, it did

not impair either the pre-acquired discrimination ability or the expression of CR itself after reaching an asymptotic level of learning. These results were almost similar to those with the non-selective mAChR antagonist scopolamine [19], suggesting that among the several types of mAChRs the M1 receptors may play an important role in the formation of the conditional discrimination memory.

### 4.1. Involvement of the M1 mAChRs in the acquisition of the discriminative eyeblink CRs

The present results suggest that that the M1 mAChRs play an important role in the acquisition of the discriminative eyeblink CRs during the serial feature positive discrimination task, in which participation of the hippocampus [18], as well as the mAChRs [19], has been suggested. However, several studies using the M1 mAChR deficient mice [35, 57] and VU0255035-treated rats [56] demonstrated that the M1 mAChRs are not necessarily required for some kinds of hippocampus-dependent tasks, such as Morris water maze and contextual fear conditioning. Therefore, M1 mAChRs might not directly participate in the hippocampal processing during the discriminative eyeblink CRs. In addition, there might be some contribution of extrahippocampal structures, such as the basal forebrain, which express the M1 mAChRs [41] and have close interaction with the hippocampus [5]. Further experiments using other types of M1 antagonists together with local administration to the hippocampus will be needed.

### 4.2. Impairment of overall CRs by blocking the M1 mAChRs

Different from our previous results using scopolamine [19] the selective blockade of M1 mAChRs using VU0255035 impaired the overall frequency of CRs that was calculated irrespective of the trial types (Fig 2C). Because the patients with medial temporal lobe amnesia showed impairment in discrimination but not in the overall acquisition of the CRs in the eyeblink conditional discrimination [24], the effect of selective M1 blockade with VU0255035 might not be related to the role of the hippocampus in this discrimination task, suggesting a contribution of other areas, particularly the basal forebrain, which express the M1 mAChRs [41].

There are two possible causes for this impairment. One is a direct inhibitory effect on the CR expression. But this might not be the case, because the administration of VU0255035 after sufficient learning in control mice did not impair expression of the pre-acquired CRs in cued trials as well as the discrimination between the cued and non-cued trials (Figs 2Aa and 3Aa), showing no impairment in overall CRs (Fig 2C). Another possibility is an indirect effect of the decreased contingency between the CS and US on the learning rate, caused by an inability to discriminate the CSs in the different temporal contexts during the conditioning. If an animal could not differentiate the non-cued CS-alone trials from the cued paired trials, the contingency between the CS and US would decline to 50% for that animal, which might lead to a decrease in the acquisition of the CR due to the lower contingency as reported in rabbits [58, 59] and humans [60].

### 4.3. Impairment of the anticipatory temporal pattern of CRs by blocking the M1 mAChRs

The control mice exhibited a significantly shorter onset latency of the CR in the cued trials than the non-cued trials (Fig 4A), while there was no significant difference in the peak amplitude of CR between the cued and non-cued trials (Fig 4C). Similar results and tendencies were observed in our previous report (Fig 4A in [19]) and in human conditional discrimination

eyeblink conditioning [24]. These observations suggest that the preceding conditional cue might affect the initiation of the CR by preparing the subjects for the coming CS. Similar modulatory effects of attention and anticipatory anxiety on the latency of acoustic startle eyeblink reflex have been reported in humans [61–63].

In contrast, the VU0255035-treated mice did not show a significant difference in the onset latency of CR between the cued and non-cued trials (Fig 5A). This effect of VU0255035 on the temporal pattern of CR was similar to that of scopolamine (Rahman et al., 2016), whereas the administration of scopolamine did not affect the CR onset latency in the simple delay eyeblink conditioning in mice [17]. Therefore, extinction of the difference in onset latency by blocking the mAChRs might reflect an impairment in preparing for the coming reinforced CS using the preceding light cue, associated with the impairment in differential CR% (Fig 2B). Similarly, amnesic patients with bilateral medial temporal lobe damage showed the identical onset latency of CR in the reinforced and the non-reinforced trials along with the impairment in discrimination during the eyeblink conditional discrimination task [24]. Consistent with these conditional discrimination tasks using a single tone, the rabbits with bilateral hippocampectomy showed almost the same CR onset latencies to the reinforced and the non-reinforced tone CSs during the two-tone discrimination reversal, in which they showed a great impairment [64]. Therefore, the M1 mAChRs in the higher brain regions, including the hippocampus, may play an important role in top-down modulation of the temporal pattern of sensorimotor reflex in the relatively complex discrimination tasks.

### 4.4. Possible mechanism of M1 mAChRs during learning of discriminative conditioned response

It has been suggested that the intrinsic neuronal excitability of hippocampal pyramidal neurons in the dorsal CA1 region plays a vital role in successful learning in hippocampus-dependent eyeblink conditioning. A direct involvement of the small conductance potassium (SK) channel in the trace eyeblink conditioning was shown in one study [65]. On the other hand, it was revealed that stimulation of the mAChRs in cortical pyramidal neurons promotes the intrinsic plasticity by inhibiting the SK channel [66, 67], suggesting that mAChR–SK channel interaction might play a vital role in successful learning in the trace eyeblink conditioning. Therefore, the present inhibitory effects of the M1 antagonist on the acquisition of differential CRs between the cued and non-cued trials as well as on the overall CRs, but not on the expression, during the serial feature-positive discrimination task might be related to the M1–SK interlink occurring in the higher brain region, including the hippocampus.

### 5. Conclusions

We have shown that systemic administration of the M1 mAChR antagonist VU0255035 impaired the acquisition of differential CRs between the cued and non-cued trials as well as the overall CRs during the serial feature-positive discrimination task of eyeblink conditioning in mice. However, it did not impair either the pre-acquired discrimination ability or the expression of CR itself after reaching an asymptotic level of learning. These results suggested that M1 receptors may play an important role in the formation of memory for conditional discrimination.

### Author Contributions

**Conceptualization:** Md Ashrafur Rahman, Norifumi Tanaka, Shigenori Kawahara.

**Data curation:** Md Ashrafur Rahman, Norifumi Tanaka, Md. Nuruzzaman.

**Formal analysis:** Md Ashrafur Rahman, Norifumi Tanaka, Shandhya DebNath, Shigenori Kawahara.

**Investigation:** Md Ashrafur Rahman, Shigenori Kawahara.

**Methodology:** Md Ashrafur Rahman, Norifumi Tanaka, Shigenori Kawahara.

**Supervision:** Shigenori Kawahara.

**Writing – original draft:** Md Ashrafur Rahman, Md. Nuruzzaman, Shandhya DebNath, Shigenori Kawahara.

**Writing – review & editing:** Md Ashrafur Rahman, Shigenori Kawahara.

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
