## [Decision Letter · Decision Letter 0]

9 Apr 2020

PONE-D-20-04932

Blockade of the M1 Muscarinic Acetylcholine Receptors Impairs Eyeblink Serial Feature-Positive Discrimination Learning in Mice

PLOS ONE

Dear Dr. Rahman,

Thank you for submitting your manuscript to PLOS ONE. After careful consideration, we feel that it has merit but does not fully meet PLOS ONE’s publication criteria as it currently stands. Therefore, we invite you to submit a revised version of the manuscript that addresses the points raised during the review process.

Two experts in the field have carefully evaluated the manuscript entitled,’ Blockade of the M1 Muscarinic Acetylcholine Receptors Impairs Eyeblink Serial Feature-Positive Discrimination Learning in Mice’. Their comments are appended below. Both reviewers are generally positive for publication, however Dr. Ashrafur Rahman should be taken care of various concerns mentioned below before publication. 

Since both reviewers pointed out that the authors can’t exclude the involvement of other mAChR, I recommend making further description. 

These concerns are surely strengthen the manuscript, I look forward to hearing the authors reply.

We would appreciate receiving your revised manuscript by May 24 2020 11:59PM. To enhance the reproducibility of your results, we recommend that if applicable you deposit your laboratory protocols in protocols.io, where a protocol can be assigned its own identifier (DOI) such that it can be cited independently in the future. For instructions see: http://journals.plos.org/plosone/s/submission-guidelines#loc-laboratory-protocols

We look forward to receiving your revised manuscript.

Kind regards,

Manabu Sakakibara, Ph.D.

Academic Editor

PLOS ONE

Journal Requirements:

3. Please include a copy of Table 3 which you refer to in your text on page 17.

Reviewers' comments:

Reviewer's Responses to Questions

**Comments to the Author**

1. Is the manuscript technically sound, and do the data support the conclusions?

Reviewer #1: No

Reviewer #2: Yes

2. Has the statistical analysis been performed appropriately and rigorously? 

Reviewer #1: Yes

Reviewer #2: Yes

3. Have the authors made all data underlying the findings in their manuscript fully available?

Reviewer #1: Yes

Reviewer #2: Yes

4. Is the manuscript presented in an intelligible fashion and written in standard English?

Reviewer #1: No

Reviewer #2: Yes

5. Review Comments to the Author

Reviewer #1: This experiment utilizes systemic administration of a selective M1 antagonist to assess the receptor's involvement in a serial feature-positive classical conditioning paradigm. Using a crossover drug design, these experiments find that the M1 antagonist impairs acquisition of the conditioned response, but not the expression of a previously learned conditioned response. The experiments are well designed and executed, though other factors diminish the impact of this submission. Given that the non-selective antagonist scopolamine has produced similar results, that an antagonist of one subtype of muscarinic is sufficient to do the same is not particularly surprising or impactful without other evidence of a dissociation from other muscarinic subtypes or regional specificity of effects. These studies effectively summarize that one of the five subtypes is involved in this behavior, which is not particularly noteworthy in isolation. Other specific comments follow.

It would be helpful to contextualize these studies in terms of how modulating eye-blink conditioning might pertain to clinical features or other human relevant applications.

These experiments were performed for 60-70 minutes for 10 days during the animal's inactive period when most consolidation takes place. Chronic sleep deprivation can impair consolidation of hippocampal dependent conditioning. While a non-drug control is used and the degree of sleep disruption is mild-moderate, application of findings may be limited by a lack of an active period control where sleep patterns have not been chronically disrupted.

Clarity of the results could be enhanced by grouping the figures together to reflect how they are presented in the results section or vice-versa. Jumping between figures to compare results between experimental groups is inefficient for the reader's understanding of the results. In other words, each section of the results 3.1-3.4 requires jumping from figure set 2 through 5 rather than each results section corresponding to it's own figure. Comparisons across figures and between individual experiments would be better saved for the discussion section. While this point is largely stylistic, it will greatly enhance readers' understanding of the multiple ways these results are parsed.

The present study did not provide any evidence of hippocampal dependency or independence, so it is unclear why discussion section 4.1 claims to dissociate their findings from "hippocampus-dependent" conditioning or other hippocampus-dependent learning tasks, especially as they cite that serial feature positive discrimination likely involves the hippocampus. The evidence is not compelling to support this distinction from other hippocampus dependent tasks as presented in this data or text. These conclusions reach beyond the scope of the results and inflate the purported impact of the results.

Discussion section 4.2 goes to great lengths to make the claim that extra-hippocampal M1 activity is involved in acquisition of CRs. This is not particularly controversial, especially as M1 is widely expressed throughout the basal forebrain. But it is unclear how these data are particularly relevant to that point.

Figure 1B needs axis labels for the reader to understand the timecourse and magnitide of the EMG response. In general, all of the figures could use additional annotation pertaining to what each graph is showing and which groups belong to each respective symbol. It is difficult to understand this information as it is presented in the figure legend text, but could be facilitated with additional annotation.

There are a few specific grammatical points:

Line 381: Hippocampal function is not "damaged" by scopolamine, rather it is acutely impaired.

Line 72-73: Sentence fragment, should be separated from following sentence with a comma.

Line 148: "band-path filtered" should likely be band-pass filtered

Line 206-207: "showed a moderate learning", should read showed moderate learning. No 'a' required.

Reviewer #2: This is an interesting set of data, showing that M1 muscarinic acetylcholine receptors (mAChRs) are essential for successful performance of a cued, temporal order conditioning task in mice. The study is somewhat limited in scope and a full discussion is missing. Below are some suggestions that may raise the impact of the paper.

The authors have previously shown that mAChR blockade using the non-selective antagonist scopolamine impairs this type of learning as well. The impact of the present study would be significantly stronger if the authors not only tested for the involvement of M1 receptors, but also other mAChR subtypes.

If the authors choose to not expand their study to include other receptor subtypes, they need to at least add more detail on the M1 blockade used here. The use of a pharmacological, rather than a genetic approach, and only one drug, weakens the study. The authors need to comment on the selectivity of VU0255035, and at the very least discuss the limited approach as a serious caveat.

The absence of a discussion is not beneficial to the paper. One aspect to be discussed for sure is the mechanism by which mAChRs might act. One possible path to develop this discussion is to point to prior work from the lab of John Disterhoft, showing that trace eyeblink conditioning is modulated by K+ channels of the SK-type (McKay et al., J. Neurophysiol. 108, 2012). These appear to be linked to muscarinic AChR signaling, as recently demonstrated in vitro in cortical pyramidal cells (Gill and Hansel, eNeuro 7, 2020). This mAChR – SK interaction might be crucially involved here and should be discussed.

Figures 2-5: It is hard to understand these figures without reading the legends (which is sometimes needed when figures are used in talk presentations). The authors might want to additionally explain the symbols within the figures themselves.

6. PLOS authors have the option to publish the peer review history of their article (what does this mean?). If published, this will include your full peer review and any attached files.

Reviewer #1: No

Reviewer #2: No

---

## [Author Response · Author response to Decision Letter 0]

22 May 2020

Response to Reviewers

We thank the reviewers for reading the manuscript carefully and providing productive comments. We have subsequently revised our manuscript titled "Blockade of the M1 Muscarinic Acetylcholine Receptors Impairs Eyeblink Serial Feature-Positive Discrimination Learning in Mice.”

The comments made by each reviewer are first copied below (underlined) and followed by our responses. Each line number indicated with corresponding page number in our reply refers to that in the ‘Revised Manuscript with Track Changes’, in which the lines are numbered throughout the manuscript.

1. Since both reviewers pointed out that the authors can’t exclude the involvement of other mAChR, I recommend making further description. These concerns are surely strengthen the manuscript, I look forward to hearing the authors reply.

Response: We thank for the careful reading of our manuscript.

In response to this comment, we have changes the sentences and added some lines in the introduction part (lines 77-93 in page 4) as below

Among them, the M1 receptors are most abundantly expressed in the hippocampus, constituting 40‒50% of the total mAChRs [26-29]. Studies showed that the M1 receptors are more related to memory functions compared to other mAChRs receptors. [30-31]. Besides, the M1 receptors are important for attention and memory in several learning tasks, such as passive avoidance [32-33], contextual fear conditioning [34], radial arm maze [35], T-maze [36] and considered as a potential target for memory functions [37]. The M2 receptors are also densely expressed in the hippocampus and neocortex. They play a vital role in the inhibitory modulation at dopaminergic terminals [38-40] as well as in a general anti-nociception at the spinal cord [41], but have a minor role in learning and memory compared to M1 [42]. The M3 receptors are found in the brain but at a lower level than other subtypes. They are mostly involved in regulating food intake [43] and body growth [44]. The M4 receptors are largely expressed in the corpus striatum and considered as a promising target for treating schizophrenia [45] and locomotor dysfunction such as Parkinson's disease [46-47].The M5 receptors are predominantly expressed in the pars compacta of substantia nigra and are the potential target for the treatment of drug addiction [48]. Therefore, we focused on the M1 receptors as the first candidate that plays an important role in the serial feature-positive discrimination task of eyeblink conditioning. 

Journal Requirements:

Please ensure that you have an ORCID iD and that it is validated in Editorial Manager. To do this, go to ‘Update my Information’ (in the upper left-hand corner of the main menu), and click on the Fetch/Validate link next to the ORCID field. 

Response: Yes I have an ORCID id and I validate it.

Please include a copy of Table 3 which you refer to in your text on page 17.

Response: We have removed the word “table 3” (line 412 in page 18). 

5.Review Comments to the Author

Reviewer #1: 

This experiment utilizes systemic administration of a selective M1 antagonist to assess the receptor's involvement in a serial feature-positive classical conditioning paradigm. Using a crossover drug design, these experiments find that the M1 antagonist impairs acquisition of the conditioned response, but not the expression of a previously learned conditioned response. The experiments are well designed and executed, though other factors diminish the impact of this submission. Given that the non-selective antagonist scopolamine has produced similar results that an antagonist of one subtype of muscarinic is sufficient to do the same is not particularly surprising or impactful without other evidence of a dissociation from other muscarinic subtypes or regional specificity of effects.

Response: We thank for the careful reading. In response to this comment, we have changed the sentences and added some lines in the introduction part (lines 77-93 in page 4) as below

Among them, the M1 receptors are most abundantly expressed in the hippocampus, constituting 40‒50% of the total mAChRs [26-29]. Studies showed that the M1 receptors are more related to memory functions compared to other mAChRs receptors. [30-31]. Besides, the M1 receptors are important for attention and memory in several learning tasks, such as passive avoidance [32-33], contextual fear conditioning [34], radial arm maze [35], T-maze [36] and considered as a potential target for memory functions [37]. The M2 receptors are also densely expressed in the hippocampus and neocortex. They play a vital role in the inhibitory modulation at dopaminergic terminals [38-40] as well as in a general anti-nociception at the spinal cord [41], but have a minor role in learning and memory compared to M1 [42]. The M3 receptors are found in the brain but at a lower level than other subtypes. They are mostly involved in regulating food intake [43] and body growth [44]. The M4 receptors are largely expressed in the corpus striatum and considered as a promising target for treating schizophrenia [45] and locomotor dysfunction such as Parkinson's disease [46-47].The M5 receptors are predominantly expressed in the pars compacta of substantia nigra and are the potential target for the treatment of drug addiction [48]. Therefore, we focused on the M1 receptors as the first candidate that plays an important role in the serial feature-positive discrimination task of eyeblink conditioning. 

It would be helpful to contextualize these studies in terms of how modulating eye-blink conditioning might pertain to clinical features or other human relevant applications.

Response: We thank again for the careful reading. We have pointed out some lines which relate the eyeblink conditioning with the clinical features on human. (lines 73-75 in page 4) and (lines 376-378 in page 16).

These experiments were performed for 60-70 minutes for 10 days during the animal's inactive period when most consolidation takes place. Chronic sleep deprivation can impair consolidation of hippocampal dependent conditioning. While a non-drug control is used and the degree of sleep disruption is mild-moderate, application of findings may be limited by a lack of an active period control where sleep patterns have not been chronically disrupted.

Response: We thank again for the careful reading. As pointed out, there was a study that showed impairments in memory consolidation by sleep deprivation “after” the daily eyeblink conditioning (De Zeeuw and Canto, 2020). Although there might be concerns about any adverse effects of sleep interference “during” the eyeblink conditioning, it was shown that there was no difference in memory acquisition or learning rate between the rats conditioned in the light period and those conditioned in the dark period, even in the hippocampus-dependent trace eyeblink conditioning (Weiss et al., 1999).

De Zeeuw CI and Canto CB (2020) Sleep deprivation directly following eyeblink-conditioning impairs memory consolidation. Neurobiology of Learning and Memory, 170, 107165

Weiss C et al. (1999) Trace eyeblink conditioning in the freely moving rat: Optimizing the conditioning parameters. Behav Neurosci., 113:1100-1105.

Clarity of the results could be enhanced by grouping the figures together to reflect how they are presented in the results section or vice-versa. Jumping between figures to compare results between experimental groups is inefficient for the reader's understanding of the results. In other words, each section of the results 3.1-3.4 requires jumping from figure set 2 through 5 rather than each results section corresponding to it's own figure. Comparisons across figures and between individual experiments would be better saved for the discussion section. While this point is largely stylistic, it will greatly enhance readers' understanding of the multiple ways these results are parsed.

Response: In response to this comment, we have rearranged the result sections. The first two sections (section 3.1 and 3.2) illustrate the results of figures 2 and 3 (lines 189-289 in pages 9-13). The sections 3.3 and 3.4 describes the results of figures 4 and 5 (lines 290-341 in pages 13-15). 

The present study did not provide any evidence of hippocampal dependency or independence, so it is unclear why discussion section 4.1 claims to dissociate their findings from "hippocampus-dependent" conditioning or other hippocampus-dependent learning tasks, especially as they cite that serial feature positive discrimination likely involves the hippocampus. The evidence is not compelling to support this distinction from other hippocampus dependent tasks as presented in this data or text. These conclusions reach beyond the scope of the results and inflate the purported impact of the results. 

Response: Considering the reviewer’s comment, we removed the latter part of discussion section 4.1 to avoid over-speculation (see revised manuscript with track changes indicated by strike-through; lines 364-376 in page 16). Also we have deleted the corresponding references lines 626-629 in pages 27-28). 

Discussion section 4.2 goes to great lengths to make the claim that extra-hippocampal M1 activity is involved in acquisition of CRs. This is not particularly controversial, especially as M1 is widely expressed throughout the basal forebrain. But it is unclear how these data are particularly relevant to that point.

Response: According to the reviewer’s comment, we removed the latter part of discussion section 4.2 to avoid over-speculation (see revised manuscript with track changes indicated by strike-through; lines 395-405 in pages 17-18). Also we have deleted the corresponding references lines 641-652 in pages 28-29).

Figure 1B needs axis labels for the reader to understand the timecourse and magnitide of the EMG response. In general, all of the figures could use additional annotation pertaining to what each graph is showing and which groups belong to each respective symbol. It is difficult to understand this information as it is presented in the figure legend text, but could be facilitated with additional annotation.

Response: Thanks again for the cautious reading of our manuscript. We have made the changes by labelling the axis and incorporated the corresponding figure legend, mentioned in Page 30; lines 683-684.

In addition, we have incorporated the additional annotation of figures 2 to 5 and rearranged the sentences with adding some words in the corresponding figure legend of figure 2, mentioned in Pages 30-31; lines 696-698 and in Page 31; lines 707-708.

. 

Line 381: Hippocampal function is not "damaged" by scopolamine, rather it is acutely impaired.

Response: Thanks for the cautious reading of our manuscript. We have deleted the line (line 400 in page 17).

Line 72-73: Sentence fragment, should be separated from following sentence with a comma

Response: Thanks again for the careful reading of our manuscript. We have separated the sentences by using comma (line 74 in page 4).

Line 148: "band-path filtered" should likely be band-pass filtered

Response: We have made the changes by writing “band-pass filtered” in place of "band-path filtered" (line 161 in page 7).

Line 206-207: "showed a moderate learning", should read showed moderate learning. No 'a' required.

Response: We have made the changes by deleting ‘a’ from the "showed a moderate learning" (line 219-220 in page 10).

Reviewer#2: 

If the authors choose to not expand their study to include other receptor subtypes, they need to at least add more detail on the M1 blockade used here. The use of a pharmacological, rather than a genetic approach, and only one drug, weakens the study. The authors need to comment on the selectivity of VU0255035, and at the very least discuss the limited approach as a serious caveat. The absence of a discussion is not beneficial to the paper. One aspect to be discussed for sure is the mechanism by which mAChRs might act. 

Response: We thank to the reviewer for arising such type of important query. In 

response to this comment, we have discussed the mechanism by which M1

subtypes act in the manuscript in pages: 19-20; lines 434-447. The lines are as

below

4.4. Possible mechanism of M1 mAChRs during learning of discriminative conditioned response

It has been suggested that the intrinsic neuronal excitability of hippocampal pyramidal neurons in the dorsal CA1 region plays a vital role in successful learning in hippocampus-dependent eyeblink conditioning. A direct involvement of the small conductance potassium (SK) channel in the trace eyeblink conditioning was shown in one study [60]. On the other hand, it was revealed that stimulation of the mAChRs in cortical pyramidal neurons promotes the intrinsic plasticity by inhibiting the SK channel [61, 62], suggesting that mAChR‒SK channel interaction might play a vital role in successful learning in the trace eyeblink conditioning. Therefore, the present inhibitory effects of the M1 antagonist on the acquisition of differential CRs between the cued and non-cued trials as well as on the overall CRs, but not on the expression, during the serial feature-positive discrimination task might be related to the M1‒SK interlink occurring in the higher brain region, including the hippocampus.

References

60. McKay BM, Oh MM, Galvez R, Burgdorf J, Kroes RA, Weiss C, et al. Increasing SK2 channel activity impairs associative learning. Journal of Neurophysiology. 2012; 108(3): 863‒70.

61. Gill DF, Hansel C. Muscarinic modulation of SK2-type K+ channels promotes intrinsic plasticity in L2/3 pyramidal neurons of the mouse primary somatosensory cortex. Eneuro. 2020; 0453‒19.

62. Giessel AJ, Sabatini BL. M1 muscarinic receptors boost synaptic potentials and calcium influx in dendritic spines by inhibiting postsynaptic SK channels. Neuron. 2010; 68(5): 936‒47.

Figures 2-5: It is hard to understand these figures without reading the legends (which is sometimes needed when figures are used in talk presentations). The authors might want to additionally explain the symbols within the figures themselves.

Response: Thanks again for the careful reading of our manuscript. We have made the changes by incorporating the additional annotation of figures 2-5 and rearranged the sentences with adding some words in the corresponding figure legend of figure 2, mentioned in Pages 30-31; lines 696-698 and in Page 31; lines 707-708.

---

## [Decision Letter · Decision Letter 1]

3 Jun 2020

PONE-D-20-04932R1

Blockade of the M1 Muscarinic Acetylcholine Receptors Impairs Eyeblink Serial Feature-Positive Discrimination Learning in Mice

PLOS ONE

Dear Dr. Rahman,

Thank you for submitting your manuscript to PLOS ONE. After careful consideration, we feel that it has merit but does not fully meet PLOS ONE’s publication criteria as it currently stands. Therefore, we invite you to submit a revised version of the manuscript that addresses the points raised during the review process.

The revised manuscript was carefully reviewed by the two original referees. The second one satisfied with your revision, while the first one criticized the unsolved several important concerns. 

I would like you to respond these concerns which will surely strengthen your manuscript.

We look forward to receiving your revised manuscript.

Kind regards,

Manabu Sakakibara, Ph.D.

Academic Editor

PLOS ONE

Reviewers' comments:

Reviewer's Responses to Questions

**Comments to the Author**

1. If the authors have adequately addressed your comments raised in a previous round of review and you feel that this manuscript is now acceptable for publication, you may indicate that here to bypass the “Comments to the Author” section, enter your conflict of interest statement in the “Confidential to Editor” section, and submit your "Accept" recommendation.

Reviewer #1: All comments have been addressed

Reviewer #2: All comments have been addressed

2. Is the manuscript technically sound, and do the data support the conclusions?

Reviewer #1: Yes

Reviewer #2: Yes

3. Has the statistical analysis been performed appropriately and rigorously? 

Reviewer #1: Yes

Reviewer #2: Yes

4. Have the authors made all data underlying the findings in their manuscript fully available?

Reviewer #1: Yes

Reviewer #2: Yes

5. Is the manuscript presented in an intelligible fashion and written in standard English?

Reviewer #1: Yes

Reviewer #2: Yes

6. Review Comments to the Author

Reviewer #1: The authors have been sufficiently receptive and responsive to specific comments from the initial review. Their paper is technically sound and the experiments are well thought out. However, reservations regarding the impact of these studies in isolation persist. While the authors correctly point out that 40-50% of muscarinic receptors are M1, there are 50-60% of muscarinic receptors that are not M1. Also, while M2 only plays a minor role in this type of memory, it still serves some role and could produce similar effects. Further, while this is a hippocampal dependent task, that necessarily means that this task is also dependent on the integrity of afferent inputs to hippocampus, particularly from other basal forebrain structures where M1 receptors are expressed. While the authors have appropriately tempered the scope of their findings, systemic administration of a selective agonist absent dissociation from other muscarinc receptors, and without any local specificity does not appreciably build on current literature.

Reviewer #2: (No Response)

7. PLOS authors have the option to publish the peer review history of their article (what does this mean?). If published, this will include your full peer review and any attached files.

Reviewer #1: No

Reviewer #2: No

---

## [Author Response · Author response to Decision Letter 1]

16 Jul 2020

Response to Reviewers

We thank the reviewers for reading the manuscript carefully and providing productive comment. We have subsequently revised our manuscript titled "Blockade of the M1 Muscarinic Acetylcholine Receptors Impairs Eyeblink Serial Feature-Positive Discrimination Learning in Mice.”

The comments made by each reviewer are first copied below (underlined) and followed by our responses. Each line number indicated with corresponding page number in our reply refers to that in the ‘Revised Manuscript with Track Changes’, in which the lines are numbered throughout the manuscript.

Reviewers' comments:

Reviewer's Responses to Questions

Comments to the Author

1. If the authors have adequately addressed your comments raised in a previous round of review and you feel that this manuscript is now acceptable for publication, you may indicate that here to bypass the “Comments to the Author” section, enter your conflict of interest statement in the “Confidential to Editor” section, and submit your "Accept" recommendation.

Reviewer#1: All comments have been addressed

Reviewer #2: All comments have been addressed

Response: We thank for the careful reading of our manuscript.

2. Is the manuscript technically sound, and do the data support the conclusions?

Reviewer#1: Yes

Reviewer #2: Yes

Response: We thank for the cautious reading of our manuscript.

3. Has the statistical analysis been performed appropriately and rigorously?

Reviewer#1: Yes

Reviewer #2: Yes

Response: We again thank for the careful reading of our manuscript.

4. Have the authors made all data underlying the findings in their manuscript fully available?

Reviewer#1: Yes

Reviewer #2: Yes

Response: We acknowledge for the careful reading of our manuscript.

5. Is the manuscript presented in an intelligible fashion and written in standard English?

Reviewer#1: Yes

Reviewer #2: Yes

Response: We thank for the careful reading of our manuscript.

5.Review Comments to the Author

Reviewer #1: 

The authors have been sufficiently receptive and responsive to specific comments from the initial review. Their paper is technically sound and the experiments are well thought out. However, reservations regarding the impact of these studies in isolation persist. While the authors correctly point out that 40-50% of muscarinic receptors are M1, there are 50-60% of muscarinic receptors that are not M1. Also, while M2 only plays a minor role in this type of memory, it still serves some role and could produce similar effects. 

Further, while this is a hippocampal dependent task, that necessarily means that this task is also dependent on the integrity of afferent inputs to hippocampus, particularly from other basal forebrain structures where M1 receptors are expressed. While the authors have appropriately tempered the scope of their findings, systemic administration of a selective agonist absent dissociation from other muscarinc receptors, and without any local specificity does not appreciably build on current literature.

Response: We thank for the careful reading. In response to this comment, we have made some changes in the different parts of manuscript. 

We have added some lines (lines 81-93 in page 4, lines 96-98 in page 5, lines 106-107 in page 5) and deleted one line in the introduction part (see revised manuscript with track changes indicated by strike-through; lines 89-90 in page 4). We have added the corresponding references lines 560-572 in pages 24-25 and lines 585-587 in page 25) as below

In the hippocampus-dependent trace eyeblink conditioning it was shown that selective activation of the M1 receptors improved the memory in aged rabbits by enhancing the excitability of hippocampal pyramidal neurons [38]. Consistent with this, an electrophysiological study using mAChRs knock-out mice revealed that the M1 but not the M3 receptors are involved in the cholinergic enhancement of hippocampal long-term potentiation [39]. Immunohistochemical study also showed a selective increase in immunoreactivity of PKCγ isoform after trace eyeblink conditioning, suggesting the involvement of signaling pathways of M1 receptors [40]. In contrast to the abundance of M1 receptors in the pyramidal neurons [41] which are highly engaged in learning the hippocampus-dependent eyeblink [38], the M2 receptors are present only in the nonpyramidal neuron in the cortex and hippocampus [27]. In addition, they are densely expressed on GABAergic interneurons [42]. 

Consistent with this, a significant correlation was found between the performance in spatial learning and the M1 receptor binding, but not the M2, in the hippocampus of aged monkeys [47].

Therefore, we focused on the M1 receptors as the first candidate that plays an important role in the serial feature-positive discrimination task of eyeblink conditioning, although the other subtypes of mAChRs that constitute 50‒60% of the hippocampal mAChRs might serve some roles in this learning as well.

Also, we have added some lines (lines 373-375 in page 16), modified the sentence by using “particularly the basal forebrain, which” (line 386 in page 17) in the discussion part and “systemic administration of” (line 443 in page 19) in the conclusion part as below

In addition, there might be some contribution of extrahippocampal structures, such as the basal forebrain, which express the M1 mAChRs [41] and have close interaction with the hippocampus [5].

References

5. Mckinney MI, Coyle JT, Hedreen JC. Topographic analysis of the innervation of the rat neocortex and hippocampus by the basal forebrain cholinergic system. Journal of Comparative Neurology. 1983; 217(1): 103‒21.

27. Levey AI, Edmunds SM, Koliatsos V, Wiley RG, Heilman CJ. Expression of m1-m4 muscarinic acetylcholine receptor proteins in rat hippocampus and regulation by cholinergic innervation. Journal of Neuroscience. 1995; 15(5): 4077‒92.

38. Weiss C, Kronforst‐Collins MA, Disterhoft JF. Activity of hippocampal pyramidal neurons during trace eyeblink conditioning. Hippocampus. 1996; 6(2): 192-209.

39. Shinoe T, Matsui M, Taketo MM, Manabe T. Modulation of synaptic plasticity by physiological activation of M1 muscarinic acetylcholine receptors in the mouse hippocampus. Journal of Neuroscience. 2005; 25(48): 11194-200.

40. Van der Zee EA, Kronforst‐Collins MA, Maizels ET, Hunzicker‐Dunn M, Disterhoft JF. γisoform‐selective changes in PKC immunoreactivity after trace eyeblink conditioning in the rabbit hippocampus. Hippocampus. 1997; 7(3): 271-85.

41. Levey AI. Muscarinic acetylcholine receptor expression in memory circuits: implications for treatment of Alzheimer disease. Proceedings of the National Academy of Sciences. 1996; 93(24): 13541‒6.

42. Ballinger EC, Ananth M, Talmage DA, Role LW. Basal forebrain cholinergic circuits and signaling in cognition and cognitive decline. Neuron. 2016; 91(6): 1199-218.

47. Haley GE, Kroenke C, Schwartz D, Kohama SG, Urbanski HF, Raber J. Hippocampal M1 receptor function associated with spatial learning and memory in aged female rhesus macaques. Age. 2011; 33(3): 309-20.

Beside these modification, we have made some changes by adding “the eyeblink” in place of “an eyeblink” (line 44 in page 2), “the” and “receptors” (line 96 page 5) and “eyeblink’ (line 365 in page 16) 

Reviewer #2: (No Response)

Response: We acknowledge for the careful reading.

---

## [Decision Letter · Decision Letter 2]

28 Jul 2020

Blockade of the M1 Muscarinic Acetylcholine Receptors Impairs Eyeblink Serial Feature-Positive Discrimination Learning in Mice

PONE-D-20-04932R2

Dear Dr. Rahman,

We’re pleased to inform you that your manuscript has been judged scientifically suitable for publication and will be formally accepted for publication once it meets all outstanding technical requirements.

Kind regards,

Manabu Sakakibara, Ph.D.

Academic Editor

PLOS ONE

Additional Editor Comments (optional):

Reviewers' comments:

Reviewer's Responses to Questions

**Comments to the Author**

1. If the authors have adequately addressed your comments raised in a previous round of review and you feel that this manuscript is now acceptable for publication, you may indicate that here to bypass the “Comments to the Author” section, enter your conflict of interest statement in the “Confidential to Editor” section, and submit your "Accept" recommendation.

Reviewer #3: All comments have been addressed

2. Is the manuscript technically sound, and do the data support the conclusions?

Reviewer #3: Yes

3. Has the statistical analysis been performed appropriately and rigorously? 

Reviewer #3: Yes

4. Have the authors made all data underlying the findings in their manuscript fully available?

Reviewer #3: Yes

5. Is the manuscript presented in an intelligible fashion and written in standard English?

Reviewer #3: Yes

6. Review Comments to the Author

Reviewer #3: The authors are addressing the concerns of the reviewers, although not necessarily directly in several points.

7. PLOS authors have the option to publish the peer review history of their article (what does this mean?). If published, this will include your full peer review and any attached files.

Reviewer #3: No